# A New Developmental Approach for Judo Focusing on Health, Physical, Motor, and Educational Attributes

**DOI:** 10.3390/ijerph20032260

**Published:** 2023-01-27

**Authors:** Fernando Garbeloto, Bianca Miarka, Eduardo Guimarães, Fabio Rodrigo Ferreira Gomes, Fernando Ikeda Tagusari, Go Tani

**Affiliations:** 1Centre of Research Education, Innovation and Intervention in Sport (CIFI2D), Faculty of Sport, University of Porto, 4200-450 Porto, Portugal; 2Department of Fights, School of Physical Education and Sports, Federal University of Rio de Janeiro, Rio de Janeiro 21941-599, Brazil; 3School of Physical Education and Sport, University of São Paulo, São Paulo 05508-030, Brazil

**Keywords:** martial arts, combat sports, pedagogical process

## Abstract

Judo is currently the most practiced combat sport in the world. There is no doubt of its importance for children, adolescents, adults, and seniors. However, due to its inherent complexity, coaches tend to place greater or lesser emphasis only on one of its multiple domains during the judoka formative years. The present study presents a novel developmental approach signaling Judo as an excellent way for physical, motor, educational, and health development across the lifespan. In this study, we consulted 23 Judo coaches, aiming to clarify the relevance and adequacy of the domains (Competitive, Health, Technical-Tactical, and Philosophical-Educational) and phases of the proposed model. Our findings showed that the model contents—physical, motor, educational, and health—are entwined with its four domains, which were considered of utmost importance by experts in the development of judokas. We, therefore, contend that our model is essential to better understand the growth and development of young judokas. Also, it may be a useful tool for Motor Development experts.

## 1. Introduction

Physical inactivity and sedentariness affect thousands of children and adolescents around the world [1]. According to UNESCO [2], approximately 80% of school-age children and youngsters do not meet the daily recommendations for physical activity. In addition, the pandemic period further impacted the development of children and adolescents increasing their sedentariness and restricting physical, motor, and social development [3,4].

It is known that low levels of physical and motor performance increase youngsters’ risks of developing sedentary habits [5,6,7] and becoming sedentary adults [8]. To counterbalance this negative trend, sports participation is a powerful way of reducing sedentariness and its negative health consequences (e.g., obesity, hypertension, diabetes, psychological) [9,10].

Judo is one of the most widely practiced Olympic sports worldwide—people from four to 80+ years old participate in more than 200 countries affiliated with the International Judo Federation [11]. There is strong evidence that the systematic practice of Judo can bring numerous benefits to health [12,13], as well as the enhancement of physical [14], motor [15], social [16], and psychological [17,18] development. In addition, it promotes the inclusion of people with specific needs [19].

As it is practiced by thousands of children around the world, Judo is an excellent sport to encourage, from the first years of life, a physically active lifestyle, as well as the reduction of sedentary habits [20]. Furthermore, Judo can be practiced with different levels of intensity and different goals (e.g., competitive, recreational, or health-oriented) across the lifespan [21].

The Judo originator, Professor Jigoro Kano, conceived it as a “gentle way” to achieve physical, motor, intellectual, and educational development. Furthermore, with its inclusion in the Olympic Games in 1964, Judo was predominantly viewed as a sport combat both in society, as well as in academia [22,23]. Within the academic community, Judo was the object of research mainly focusing on competitive performance [22]. Surprisingly, very few scientific reports looked at it from an educational perspective aiming at physical, psychological, and social development combined with health benefits [24,25].

In society, the importance of competition can be represented by the several Judo national federations that have official competition categories for children starting at nine years of age, thus encouraging competitive participation in the early years. This emphasis may induce negative consequences for Judo practitioners, namely, (i) greater chances for young judoka to burnout or develop high levels of stress causing premature drop-out [26]; (ii) early specialization in few Judo techniques, restricting the possibility of enhancing childhood motor competence [27]; (iii) reduced amount of time dedicated to teaching the educational aspects of Judo; and, finally, (iv) the development of judokas mainly focused on the competitive domain, who tend to devalue the benefits related to educational, health, and well-being, i.e., they are less aware of Judo’s main goals as imagined by Professor Jigoro Kano [23].

The focus on the competitive side of Judo has other consequences in the way new teachers are trained. For example, it is known that teacher training is often carried out by a repetitive strategy; that is, a teacher who, as a student, focused only on competitive characteristics tends to replicate this in their teaching [28]. As such, Judo’s manifold benefits tend to be neglected. One way to counterbalance the excess of the competitive aspect is to employ other sporting models favoring the holistic training of the young athlete, namely, the Long-Term Athlete Development and the Youth Physical Development models [29,30]. Although these models are of great importance for the training process, none contemplates the health benefits of Judo or the philosophical and educational aspects inherent to it.

There is an evident need for a new framework that integrates the specific goals of Judo. Hence, in this study we present a novel developmental approach signaling Judo as an excellent way to achieve physical, motor, educational, and health development across the lifespan. In practical terms, we predict this approach will help Judo coaches with different levels of experience to better design their training programs considering every proposed domain. In academic terms, we expect to attain a better link between practical and academic aspects of young judoka development, as well as prompt researchers to test the main ideas of our proposal. 

## 2. Materials and Methods

### 2.1. Procedures for the Approach Development

The development and validation of a developmental approach can be achieved following a series of three steps: the first is literature-based; the second relies on the judgments of experts in the field; the third is done by coaches and researchers who test the ideas proposed in the new approach. In the present paper, however, we will only consider the first two steps.

#### 2.1.1. Literature Review

In the literature-based step, we first sought a clear understanding of Judo’s essential goals, as outlined in Professor Jigoro Kano writings [31,32,33,34]. Then, we searched for studies discussing the aims of Judo, as well as the judokas’ unfolding during the formative years [23,35,36,37,38].

#### 2.1.2. Expert Judgement

In the expert judgement, we first relied on the highly consistent knowledge from Motor Development experts [39,40,41,42,43]. The main aim was to suggest reliable instructions for Judo teachers, to help them understand which contents should be taught in each phase of judoka development, taking advantage of windows of opportunity to enhance the development of physical, cognitive, and motor domains. Secondly, we invited 23 Judo coaches with different levels of academic background, grades, and years of experience teaching judokas of different ages. The experts were invited to check whether Judo teachers with different grades and experience understood our approach and whether they wanted to introduce, modify, or remove any information. To complete this step, each expert received a link to access an online questionnaire within the Google Forms platform, containing 17 questions divided into three sections: (i) understanding of the approach; (ii) relevance of its domains; (iii) adequacy of the different phases. In each section, together with closed questions, there were open questions in which experts could change, remove, or make suggestions they considered relevant. 

This study was approved by the Research Ethics Committee of the School of Physical Education of the University of São Paulo (registration number: CAAE 46606421.3.0000.5391).

### 2.2. Developmental Approach and Structure 

After consulting the literature about Judo and Motor Development, a heuristic model was created to represent our approach. The model comprises two inverted triangles flanked by four domains—Competitive, Health, Technical, and Philosophical-Educational—which were chosen to represent the main goals of Judo (Figure 1). At the end of these domains, there are positive and negative signs representing the emphasis that the teacher must attach to each domain at each stage of the judoka development.

It is worth clarifying that the negative sign does not indicate the absence of stimuli in a given domain. On the contrary, the sign indicates that the domain should receive greater (+) or less (−) emphasis at a given phase, i.e., all domains must be dealt with concomitantly as the judoka formation process unfolds. There are only two domains that did not receive positive or negative signs—Technical, and Philosophical-Educational domains—since these must be addressed throughout the entire process, regardless of the competitive level or the stage of development. Finally, the proposal also includes solid and dashed lines. The dashed lines indicate that the judoka can go back and forth as often as necessary from one level to the next. For example, in some cases, the judoka needs to return to the level of development of fundamental movement skills (FMS) and only later practice with Judo-specific motor skills. Please note that this can also occur between the three levels of competition considered in the approach: national, regional, and recreational. The solid lines that close the model at the national and regional levels indicate that at a given moment, for different reasons (e.g., physical, motivational), the judoka ended his/her competitive career. The recreational level does not have lines at the top because, at this non-competitive level, the Health domain is predominant, and the judoka does not have any need or obligation to compete.

#### 2.2.1. The Definition of the Domains

After a careful reading of Professor Jigoro Kano’s writings, four domains were elected to represent the goals of Judo in relation to the development of the judokas. Available evidence shows that Professor Kano formulated a training method consisting of four essentials: (i) kata: the execution of formal movement patterns that constitute ideal movement models; (ii) randori: training sessions with the opponent, in which attack and defense are mutually practiced; (iii) kogi: explanations about the principles of Judo; (iv) mondo: question and answer sessions between students and the teacher [32,38]. According to the originator, these essentials must be practiced based on two fundamental principles: minimum effort and maximum efficiency (seiryoku-zenyo) and mutual welfare and benefit (jita-kyoei) [33].

These principles were developed to be transferred to other fields of human activity. Thus, in Professor Kano’s words, “the Judo practice can improve the human body, making it strong, healthy, and useful. It can also be applied to improving intellectual and moral power, which constitutes mental and moral education” [31]. Further, according to his words, philosophical and educational principles should be applied at all times in the judoka’s life. For this reason, one of the domains chosen to represent our approach was Philosophical-Educational, in order to cover educational, social, inclusion, and philosophical themes. Professor Kano also contended that the practice of kata and randori was essential for technical and health development. Thus, two other domains were considered in our approach: Health (covering ways to combat sedentary lifestyles and enhance health and physical development themes) and Technical (covering technical and tactical themes) domains [31,32,33]. In addition, a fourth domain was included: the Competitive domain (covering different levels of competition). After the Judo inclusion in the Olympic Games, this domain became the most representative of Judo expressions for society and for judokas themselves [22].

It is important to highlight that although the four domains are presented separately, they interact and affect the entire judoka’s development. Therefore, we contend that Judo coaches must emphasize the teaching of the four domains, but with different magnitudes, in each developmental phase.

#### 2.2.2. The Phases

In addition to these domains, our approach also comprises three different phases of the judoka’s development: Formative Years, Decision, and Definitive (from the beginning of the journey until the moment of “retirement”). These phases were created based on consolidated Motor Development knowledge [39,40,41]. That is, Motor Development processes follow a hierarchically organized structure in which the experiences acquired in one phase significantly influence the development of the next phase [39,43].

##### The Formative Years

The Formative Years phase represents the fundamental ground on which every judoka unfolds. It comprises the first steps of the judoka in the dojo (the place where Judo is practiced). This phase begins around age 4 and lasts until age 15. Due to the interdependence between the phases, all the experiences accumulated during this phase will significantly impact on the developmental trajectories of the judokas. 

a.Philosophical-Educational Domain in the Formative Years

During these first years of formation, the judoka will learn about greetings, clothing, the meanings of the kanji “do” and “Ju,” as well as the principles of Seryoku-Zenyo and Jita-Kyoei. For example, those who start practicing Judo around 4 years old may have a great window of opportunity to learn Japanese words and expressions linked to the philosophy of Judo [44]. It is also during this phase that the practice of Judo can contribute to the improvement of several behaviors in children, namely, serenity, courage, efficiency problem-solving, socio-moral sensitivity, helping others, and responsibility [45]. 

From a philosophical perspective, moral development is one of the basic premises of Judo, and the principles learned at this stage must be extended to the judoka’s daily life, i.e., inside and outside the dojo. So that the practice of Judo can contribute to a more egalitarian and respectful society, the sooner the judoka incorporates the philosophical principles, the better [33].

b.Technical Domain in the Formative Years

The Technical domain is also essential in this formative phase because a preliminary training plan may negatively affect the performance of the judokas at later stages [46,47]. An important aspect to consider during this phase is the development of the fundamental movement skills (FMS). According to the different Motor Development models, FMS (e.g., running, jumping, kicking) are those skills that will be the basis for mastering more complex skills, for example, Judo skills [39,41]. Studies indicate that children who do not master the FMS will have difficulties mastering sports skills [48,49], that is, poor performance in FMS may negatively impact the judoka’s ability to learn and proficiently perform Judo-specific techniques.

To better understand the dynamics by which FMS develop towards more complex specific motor skills, two processes have been suggested: increasing diversity and complexity of behavior [42,43]. The first refers to increases in the number of behavior elements, the second to increases in the interaction between these elements. For example, the judoka acquires the FMS of turning the body on its axis (Mae-mawari-sabaki), and, based on this pattern, he or she will be able to develop diversified ways of moving to attack and defend (e.g., Mae-Sabaki, Ushiro-Sabaki), varying the movement parameters of speed, strength, and direction without losing efficiency. Once this skill is mastered, the judoka can start learning the Koshi-guruma (Judo technique) and, by the same process, diversify it, applying the same parameters in different ways [42]. In addition to the Mae-mawari-sabaki, other FMS can also be crucial for developing various Judo skills. For example, the Shintai (a set of movement techniques) side, front, and back shift form the basis for various Tachi-Waza techniques (i.e., projection skills).

Jumps and rolls are FMS for a specific takedown and defense techniques, while hopping on one leg is essential for learning different Judo skills (e.g., Harai-Goshi, O-Soto-gari). In contrast, ground shifts are the basis for Katame-waza techniques (ground techniques). Finally, all fall techniques (Ukemi) can be considered FMS, as they contribute to preventing injuries [50] and will be the basis for a set of Judo techniques (e.g., the Ma-sutemi-waza techniques). The acquisition of a broad, efficient, and varied FMS repertoire is a time-consuming process that should be encouraged throughout the formative years of the judokas. 

Regarding the tactical component during this phase, aspects related to competitive tactics should not be emphasized because competitive performance in younger categories does not predict performance in adulthood [51]. During the Formative Years, all coaches must encourage their judokas to learn techniques on both sides (right and left), as this can be a determinant of the technical and tactical level that they will achieve in the future [52].

c.Competitive and Health Domains in the Formative Years

As described in the model (Figure 1), during this phase, competition should receive little emphasis. As aforementioned, the harmful effect of early competition can lead young judokas to burnout or develop high levels of stress and, consequently, to drop out [26]. Also, early specialization in few Judo techniques limits gains in motor competence in children [27].

It is well-known that biological maturation acts as a performance confounder during athletic development. This is most probably the reason why being a successful competitor during childhood and pre-adolescence does not predict future competitive success [51]. Furthermore, a recent study revealed that athletes born in the first half of the year have a competitive advantage over athletes born in the second half of the year, with this influence being felt more in the under-15 male categories [53], as this is the moment when, on average, boys pass through the adolescent growth spurt. With this in mind, during this phase, Judo coaches should dedicate their training regiments to the development of motor competence. In fact, good levels of motor competence are directly associated with health factors, such as high aerobic resistance and low levels of fat percentage.

In addition, it is worth mentioning that the age at peak height velocity (PHV) is directly associated with two essential physical capacities for good competitive performance and health: strength and endurance [12,54,55]. Furthermore, preceding authors suggested that the period during and after the PHV is an excellent time to start placing greater emphasis on enhancing strength and endurance capabilities. It is also vital for the training of several techniques, as well as for increasing performance levels in randori [54].

##### Decision Phase

This phase corresponds to the central part of our approach. For the judokas who started practicing Judo before 10 years old, the Decision phase tends to occur between 16 and 20 years of age. In this phase, the judoka should decide at which level he or she will, or not, compete. It is highlighted that all the experiences accumulated in the previous phase will influence the development of the judoka during this period. For this reason, the center of the model is represented by a “filter” and the concept of constraint. According to Newell [40], changes in any motor action are related to the interactions between the constraints of the individual, whether structural (e.g., weight, height) or functional (e.g., motivation), the environment (e.g., type of tatami where Judo is practiced or social aspects), and the task (e.g., objective to be achieved). If one of these constraints changes, the resulting action also changes. For example, a recent study showed that elite female judokas aged 16–20 decreased their body fat percentage over time [56]. This change in structural constraint might influence their technical and tactical performance [57,58]. Another factor is the social condition. Although some judokas want to continue their competitive career, socioeconomic conditions might restrict the progression of their careers [59].

(a)Competitive and Technical Domains in the Decision Phase

During the Decision phase, the Competitive domain begins to receive greater emphasis. If a judoka is interested in pursuing a competitive career, it is during this period that she or he must start disputing a place in the national teams. For this reason, tactical aspects should be developed, namely, different gripping, attacks, groundwork, and the frequency of attack [60]. Additionally, the training of Judo-specific motor skills and specific physical abilities should also be prioritized. Regarding the physical capacities, it is crucial to develop not only different types of strength (e.g., isometric and dynamic) to improve performance in randori, but also aerobic capacity, which may help the judokas during randori, especially in the last moments of the fight [54]. 

During this phase, the judoka should also acquire a varied set of Judo techniques, which will be favorite techniques and possible ways to vary their application. As mentioned before, high-level judokas tend to possess more techniques and more significant variation in applying the same technique [46,61]. They explore different techniques and possibilities of applying the same technique (e.g., using different Kumi kata).

(b)Philosophical-Educational and Health Domains in the Decision Phase

If the judoka received adequate training in the previous phase, it is expected that principles, rules, and moral and ethical codes are being internalized and the judoka can use them in his daily life on and off the tatami. For example, the dynamics and different possibilities of practice make Judo a critical tool to promote the inclusion of people with specific needs. This category includes people with physical (e.g., amputated limbs, visual, hearing), motor (e.g., cerebral palsy), and intellectual (e.g., Down syndrome) disabilities. In addition to the benefits of Judo, such as the rehabilitation of balance in children with visual impairments [19], the practice between people with and without specific needs promotes inclusion and empathy among practitioners, thus promoting one of the principles fundamentals of Judo, the Jita-Kyoei.

Furthermore, during this phase, attention should also increase in the health field, especially in preventing injuries and combating sedentary lifestyles. Concerning injuries in this period, those who are, for example, between 16 and 20 years of age, despite having great physical strength and combativeness in training, might still perform some techniques less proficiently and, in doing so, increase the number of injuries [50,62]. This can not only impair competitive performance but also harm the judoka’s quality of life, depending on the level of injury. With regards to physical inactivity, the systematic practice of Judo in the form of randori, Uchikomi and the Taiso itself can be considered an essential tool in combating a sedentary lifestyle and promoting health.

##### Definitive Phase

For those who started the practice of Judo during childhood, the Definitive phase, the last of our model, should begin around the age of 20. It is necessary to clarify, however, that although our approach is subdivided by age, this does not determine the stage is the judoka is in. For example, an adult individual beginning Judo at 30 must start from the initial phase, the Formative Years. A judoka who started the practice at 7 years old may stay longer in the phase of Formative Years than a peer who started at 10 years of age. Hence, the sequence contemplated in the present approach must be respected, and the time that each judoka remains in each phase may vary. This principle follows the indications of descriptive models of Motor Development, in which the interaction between factors related to the individual (e.g., physical and motivational characteristics) and the environment (e.g., cultural context) defines how long an individual will remain in each developmental stage [39].

In this Definitive phase, the experience gathered by the judoka during the developmental process influences the competitive level that she or he must pursue from this phase on. For example, at this phase, the national level is the narrowest because, regardless of the number of judokas in a given region, the number of “vacancies” for that level will always be limited. In this case, only judokas with the best levels of performance will be able to follow this path. On the other hand, the more expansive entry, the Recreational Competitive Level, indicates that many judokas can follow this path regardless of their technical level. In this case, there is not even an obligation to compete. 

(a)Competitive and Health Domains in the Definitive Phase

At this phase, competition becomes more important in the Competitive domain since most judokas are at the senior (or adult) category, the main and higher competitive category. Thus, the judoka competes in the main competitions in national and international arenas. It is worth noting that, regardless of the level reached, over the years and with the natural decrease in physical condition, judokas tend to migrate to the Recreational level, seeking to maintain the practice of Judo focused only on their health. Here, in addition to improvements in various physical [14] and physiological capacities [12], which are essential for an active lifestyle and well-being, the systematic practice of Judo also induces gains in concentration, executive functions [17], and postural control [63]. However, care must be taken because the inappropriate intensity and volume of practice can easily cause an injury [64]. 

(b)Technical and Philosophical-Educational Domains in the Definitive Phase

In the Technical domain, regardless of the level, the judoka must always seek improvement, even though the intensity and the number of hours dedicated to the practice may vary. While a judoka at the international level trains approximately 26 h per week, 6 days per week [65], a judoka who competes at a recreational competitive level can train at an intensity and frequency that fits the reality of their day-to-day, adequately matched with their professional work, study, and family. Regarding the tactical aspects, while elite athletes try to improve tactics that favor their fighting style against the primary opponents, regional and recreational level judokas try to improve tactically against their performance. That is, the improvement must always be theirs concerning themselves.

In this phase, the Philosophical-Educational domain must be part of the judoka’s daily life. There must be a relevant dedication in transmitting the acquired knowledge to the younger judokas, mainly for those who remain in the first phase of their formative years. In addition, this sharing process can inspire older judokas to become future Judo teachers.

### 2.3. Judgment of the Experts in Judo

One hundred Judo teachers affiliated with different Brazilian Judo Confederations were invited to participate in the study. First, we invited 30 Kodanshas teachers, i.e., those with degrees above the 5th Dan of the black belt. At this stage, seven out of the 30 Kodanshas agreed to participate and answered an online questionnaire. Second, we invited 70 more experts who were below 6th Dan and had been teaching Judo for at least ten years. In total, 16 out of the 70 experts agreed to participate. On average, the Judo teachers were 44.5 ± 12.74 years old and had been teaching Judo for at least ten years (Table 1). In addition to working with children in the early stages of training, some experts had extensive international experience. For example, there is a coach of the Brazilian Judo team, an international competitor, and Faculty members who conduct research and teach Judo in their respective universities.

Along with the invitation, all teachers received a 10-page document presenting and describing, in detail, our approach, as well as a website link that gave them access to the online questionnaire. The questionnaire was created on the Google Forms platform and contained 17 questions divided into three blocks: (i) understanding of the structure of the model, (ii) domain relevance; and (iii) phase adequacy. In addition to the closed questions, in each block, there were essay questions in which the experts could change, remove, or suggest some information in the structure of the model and the content of the domains or phases.

### 2.4. Data Analysis

The descriptive statistics (percentages) for each question of the questionnaire are presented in Table 2. In order to analyze the data from the questionnaires sent to the experts, a Chi-square test was used to identify the clarity and relevance of the domains and phases included in our approach. SYSTAT 11 software was used, and the alpha level was set at 5%.

## 3. Results

Table 2 shows the tabulated results. Firstly, in the three sections of the questionnaire, none of the experts reported that the model was incomprehensible (TI), totally irrelevant (TIR), totally inadequate (TIN), incomprehensible (IC), irrelevant (IR), or inappropriate (IN). Therefore, these relative frequencies were 0%. Secondly, the experts reported that the model was comprehensible (C) or fully comprehensible (FC), relevant (R) or totally relevant (TR), and adequate (AD) or totally adequate (TA). Thirdly, no statistically significant differences between categories rated as 3 or 4 were found because associated *p*-values were higher than 5%.

In the essay questions related to the model structure, only one consideration was done by the experts, with regards to the Technical domain: (i) to integrate the tactical aspect in the title of the domain. In the other domains, experts made no suggestions. Finally, no suggestions were made in the essay questions about the Formative Years, Decision, and Definitive phases. 

## 4. Discussion

The present study aimed to present a new approach highlighting Judo as an excellent practice for physical, motor, educational, and health development across the lifespan. Given that our approach was based on Professor Jigoro Kano’s writings and models used in Motor Development research, our first concern was to examine whether Judo coaches with different levels of expertise, experience, and academic background would understand the model and find it helpful. Experts clearly understood the substantive model and its graphical representation with its inverted triangles, dotted and continuous lines, and the plus and minus signs. Their ratings reiterate this. Furthermore, no expert suggested changes or exclusions from the model. In sum, our developmental model is comprehensive for the target population and, therefore, is expected to be a helpful guide for Judo coaches with different levels of expertise.

Most of the Judo experts already had previous contact with Motor Development models or sports training methodologies in their undergraduate courses in Physical Education or training courses provided by national Judo confederations. As we were inspired by different models well-known in the available literature [39,40,41], our structure may have been easy to interpret. In addition to the Motor Development knowledge, sports literature also has many models related to the developmental trajectories of young athletes on the road to national and international success. Taken together, these models also report on the idea of windows of opportunity for young athletes’ development [29,30]. Despite their importance, current models do not address Judo-specific philosophical and educational domains, which were included in our developmental model. 

The second part of our results aimed to verify whether experts consider the four domains relevant for judokas’ unfolding across the years. Health and Philosophical-Educational domains were considered the most relevant. We contend that in the Health domain, two factors may help explain the responses given by the experts: the first is the substantial amount of information indicating that systematic participation in sports contributes to reduce the substantial number of children, youngsters, and adults with sedentary behaviors [9]; the second refers to parental concern with the health of their children. Evidence suggests that parents enroll their children in organized sports because they will tend to be physically active and that this lifestyle will be sustained in later years [66,67]. As such, Judo coaches need to be prepared to inform parents of all the Judo benefits during children’s and youngsters’ developmental processes. 

In the Philosophical-Educational domain, since Professor Jigoro Kano had advanced knowledge in the field of philosophy, we expected that the educational principles of Judo (e.g., seiryouku-zenyo and jita-kyoei) would receive the highest frequency of very relevant responses, which was the case. Evidence indicates that this domain receives little attention [22] and is generally less emphasized by Judo coaches with extensive competitive experience and minor knowledge of Judo’s benefits [28]. Because of this, our developmental model stressed the importance of promoting the Philosophical-Educational domain during the entire career of the judokas. The main goal is to incorporate the philosophical aspects of the judoka, who should employ them inside and outside the dojo.

Undoubtedly, the Competitive domain is widespread in our society and has its relevant part during the development of the judoka. Nevertheless, it received the lowest rates in the Relevant category, even though experts recognized its importance (i.e., none rated the domain as Irrelevant or Irrelevant). This finding suggests that Judo coaches are also concerned with the other domains of the sport. As such, our approach will help balance the judoka’s training process, promoting far-reaching developmental pathways in all their education and its links to health.

Finally, the Technical domain was considered highly relevant by the experts. In the document sent to them, we emphasized the importance of adequate technical development in each phase of the judoka’s development. We presented substantive evidence that the technical repertoire acquired throughout their developmental process can directly affect their competitive level [46]. Also, this process is influenced by the relationship between FMS and sports skills [47,48]. This information impacted the experts’ responses, who considered this domain extremely relevant.

In the open questions part of the document, the Technical domain was the only one that received a suggestion from one of the experts; they suggested entitling the domain the Technical-Tactical domain. Since the Technical domain also covers tactical aspects, we added the term “Tactical”. Thus, the domain is now named Technical-Tactical (see Figure 2). We believe this inclusion makes the model more comprehensive for Judo coaches. Furthermore, this may help coaches differentiate between technical and tactical aspects.

In addition to these domains, experts were invited to comment on the adequacy of each phase considered in our model. In general, all phases (Formative Years, Decision phase, and Definitive phase) received a higher frequency of responses as Totally adequate. The accumulated experience of the experts in teaching Judo may explain this finding since it is known that coaches are strongly influenced by their experience in practice [28]. These experts connected their experiences as Judo teachers with the descriptions and suggestions presented in our developmental model. For example, many coaches must have already witnessed students who stopped competing due to burnout, migrated from the national competitive level to the recreational level, or specialized too early in a single technique and did not improve their technical repertoire in the adult category.

In summary, although Judo is one of the most practiced sports in the world, little attention has been given to the conception of approaches that could help teachers to promote the goals of Judo [24,25]. In this sense, when we consider that Judo teachers tend to replicate the teaching they were given [28], the chance of favoring the competitive domain to the detriment of other domains during the judoka training process increases. The current proposal aims to encourage Judo coaches with different levels of experience to take advantage of the maximum potential that this sport has to offer for the formation of human beings. For this reason, based on the historical and recent Judo and Motor Development literature, we contend that this model can be of great importance for professionals who work with Judo at any competitive level.

## 5. Conclusions

This study presents a suitable and comprehensive model, revised and supported by experts in Judo, for the development of judokas across the lifespan. Our findings revealed that the model contents—physical, motor, educational, and health—are entwined with its four domains, which the experts in Judo considered of utmost importance. Also, this study improves our understanding of career development and sport participation in Judo. We anticipate its use for integrating Motor Development knowledge with Judo educational, philosophical, and competition values in children, youth, adults, and seniors. Once validated by experts, we recommend that this approach be tested. As such, future studies on the judokas’ unfolding careers should investigate how the physical, motor, health, and educational factors are best entwined across distinct age categories and goals of Judo.

## Figures and Tables

**Figure 1 ijerph-20-02260-f001:**
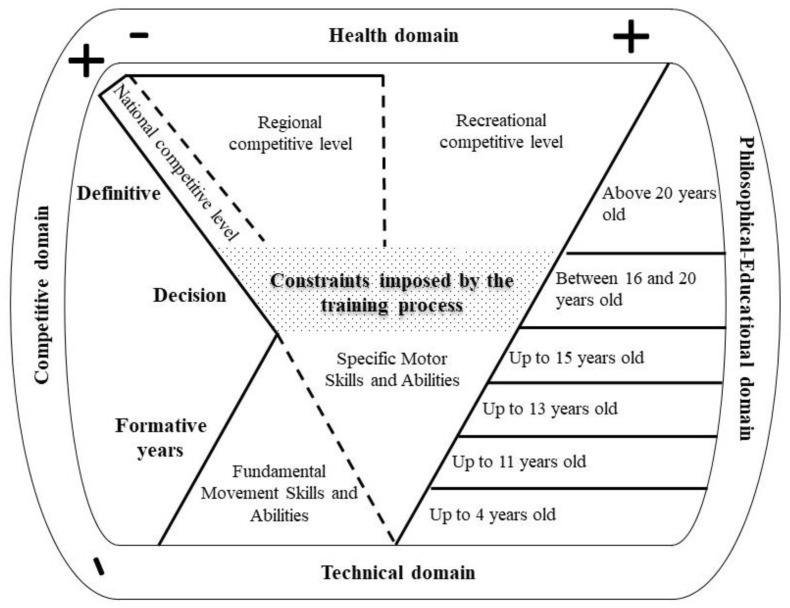
Graphical representation of the proposed model for the development of judokas throughout life, with its two triangles, four domains, and three phases.

**Figure 2 ijerph-20-02260-f002:**
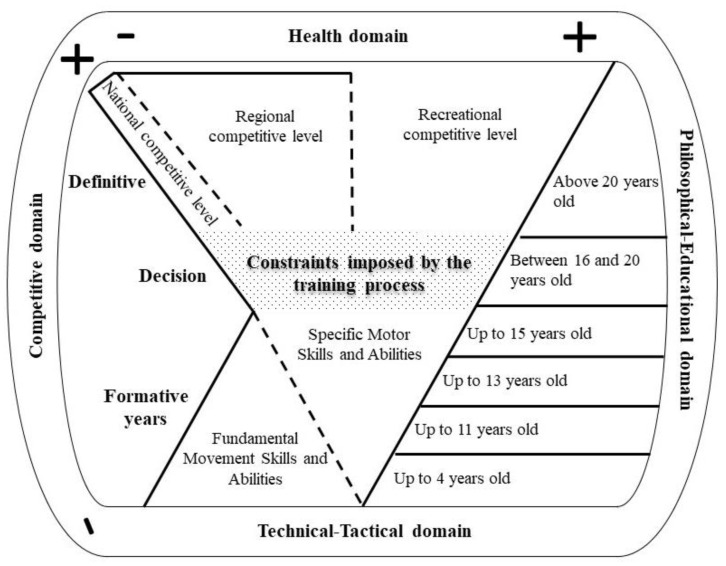
Graphical representation of the proposed model for the development of judokas throughout life, considering the responses given by the experts.

**Table 1 ijerph-20-02260-t001:** Frequency of Dan grades per experts, as well as academic degrees.

Black Belt Dan Grades						
1° Dan	2° Dan	3° Dan	4° Dan	5° Dan	6° Dan	7° Dan
*n* = 1	*n* = 6	*n* = 6	*n* = 2	*n* = 1	*n* = 4	*n* = 3
Academic Degrees						
High School Completed		Undergraduate Degree in Physical Education	Master Degree in Physical Education		PhD in Physical Education	
*n* = 5		*n* = 11	*n* = 4		*n* = 3	

**Table 2 ijerph-20-02260-t002:** Responses by the experts in Judo (percentages), chi-square results and respective *p*-values.

Questions	Percentages					
Approach comprehensiveness	TI	IC	C	FC	χ^2^	*p*-Value
Considering the approach we sent to you, do you consider its structure to be?	0%	0%	64%	36%	2.13	0.14
Relevance of the Domains	TIR	IR	R	TR	χ^2^	*p*-value
What is the relevance of the Technical domain in the judoka’s developmental process?	0%	0%	45%	55%	0.04	0.83
What is the relevance of the Philosophical-Educational domain in the judoka’s developmental process?	0%	0%	36%	64%	2.13	0.14
What is the relevance of the Competitive domain in the judoka’s developmental process?	0%	0%	55%	45%	0.39	0.53
What is the relevance of the Health domain in the judoka’s developmental process?	0%	0%	40%	60%	1.08	0.29
Adequacy of the Phases	TIN	IN	AD	TA	χ^2^	*p*-value
How suitable do you consider the Formative Years phase to be for the judoka’s developmental process?	0%	0%	31%	69%	3.52	0.06
How suitable do you consider the Decision Phase to be for the judoka’s developmental process?	0%	0%	45%	55%	0.43	0.83
How suitable do you consider the Definitive Phase to be for the judoka’s developmental process?	0%	0%	36%	64%	1.08	0.29

Notes: TI = Totally incomprehensible; IC = Incomprehensible; C = Comprehensible; FC = Fully comprehensible; TIR = Totally irrelevant; IR = Irrelevant; R = Relevant; TR = Totally relevant; TIN = Totally inadequate; IN = Inappropriate; AD = Adequate; TA = Totally adequate.

## Data Availability

Data from the present study can be obtained through the email fegarbelot@gmail.com.

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
