# Peer review of "A New Developmental Approach for Judo Focusing on Health, Physical, Motor, and Educational Attributes"

_ijerph, 2023, doi:10.3390/ijerph20032260_

Round 1
Reviewer 1 Report
This paper studied the present study presents a new developmental approach linking Judo to an excellent way for physical, motor, educational and health development across the lifespan. Because there is very few scientific reports looked at it from an educational perspective aiming at physical, psychological, and social development together with its health benefits, several Judo official competition categories may induce negative consequences for Judo practitioners. And the focus on the competitive side of Judo has other consequences in the way new teachers are trained.
Major points:
1. The significance of this paper is not fully elaborated, only from a practical point of view, there is no more literature support, that is literature is insufficient. The author need to highlight this paper's innovative contributions.
2. The main content of the article is he development and validation of our developmental approach, which was based on the literature-based and the experts’ judgments, the materials and methods are detailed and professional, especially structure and phases.The data analysis is a bit simple, but results, discussion and conclusions are specified and persuasive. Let the reader clearly understand the research process and significance of this article.
Minor points:
The picture is clear and simple, the author could add more to make it exquisite.
Author Response
Reviewer: 1
This paper studied the present study presents a new developmental approach linking Judo to an excellent way for physical, motor, educational and health development across the lifespan. Because there is very few scientific reports looked at it from an educational perspective aiming at physical, psychological, and social development together with its health benefits, several Judo official competition categories may induce negative consequences for Judo practitioners. And the focus on the competitive side of Judo has other consequences in the way new teachers are trained.
Major points:
- The significance of this paper is not fully elaborated, only from a practical point of view, there is no more literature support, that is literature is insufficient. The author need to highlight this paper's innovative contributions.
Authors’ Answer: Thank you for this suggestion. We tried highlighting the article's contributions to judo coaches and researchers in the field. The modifications are marked in yellow.
- The main content of the article is he development and validation of our developmental approach, which was based on the literature-based and the experts’ judgments, the materials and methods are detailed and professional, especially structure and phases. The data analysis is a bit simple, but results, discussion and conclusions are specified and persuasive. Let the reader clearly understand the research process and significance of this article.
Authors’ Answer: Thank you for this suggestion. We've reorganized a few paragraphs in the method and added information to make it clearer for readers (marked in yellow). We believe that the modifications in the method together with the modifications in the introduction and discussion can help the readers to understand the meaning of the article.
Minor points:
The picture is clear and simple, the author could add more to make it exquisite.
Authors’ Answer: Thank you for this further suggestion. In fact, we were able to make the image sharper and more adjusted, but since it was validated by specialists, we cannot change its structure.
Reviewer 2 Report
Congratulations on the manuscript entitled “A new developmental approach for Judo focusing on health, physical, motor, and educational attributes”. Undoubtedly, a theoretical reflection is provided that had not previously been considered, at least from the scientific point of view, very useful. Not only from a theoretical perspective, but also from a practical perspective. However, it is recommended to review the manuscript, since there are some minor flaws, especially in the journal's format, which make it have to be reviewed before acceptance.
Congratulations for the work done
Author Response
General
Congratulations on the manuscript entitled “A new developmental approach for Judo focusing on health, physical, motor, and educational attributes”. Undoubtedly, a theoretical reflection is provided that had not previously been considered, at least from the scientific point of view, very useful. Not only from a theoretical perspective, but also from a practical perspective. However, it is recommended to review the manuscript, since there are some minor flaws, especially in the journal's format, which make it have to be reviewed before acceptance.
Authors’ Answer: Thank you for the suggestion. We revised the article to make it suitable for the journal's format.

Reviewer 3 Report
Dear authors,
Thank you for the effort and effort you put into your research. I think that the approach you have made in your research on the branch of judo is interesting and important for the evaluation of experts. After some adjustments I will give below, I think your research is suitable for publication on ijerph.
Introduction
I think that this section has been written adequately, there is no need for any revision.
Methods
The Method section is really hard to read so I'll suggest some edits.
-Please give a title to the part that starts with line 78.
- It is very difficult to read and understand the "The Formative Years" part starting from Line 160, please try to explain this part by dividing it into simpler and sub-titles (it is not mandatory, but it will be easy for the readers to understand). Scientists who read this part will understand what they read. However, it can be difficult to understand from the point of view of trainers.
- Likewise, I suggest you to simplify or make more descriptive parts of "Definitive phase" and "Decision phase". I really had a hard time understanding it while reading it.
Finally, I recommend that you have the writing language you used in your research edited by a native English language expert. It will become easier to read and will also become simpler.
Thank you again for your efforts and contributions to the branch of judo.
Yours sincerely
Author Response
Comments and Suggestions for Authors
Thank you for the effort and effort you put into your research. I think that the approach you have made in your research on the branch of judo is interesting and important for the evaluation of experts. After some adjustments I will give below, I think your research is suitable for publication on ijerph.
Introduction
I think that this section has been written adequately, there is no need for any revision.
Authors’ Answer: Thanks for the comment, at the request of one of the reviewers we have added an additional paragraph to the introduction (marked in yellow).
Methods
The Method section is really hard to read so I'll suggest some edits.
Authors’ Answer: Thank you for the suggestion. In fact, the method is the central part of this article, so we've reorganized some paragraphs and added information to make the method clearer and readable.
-Please give a title to the part that starts with line 78.
Authors’ Answer: We added a title as suggested.
It is very difficult to read and understand the "The Formative Years" part starting from Line 160, please try to explain this part by dividing it into simpler and sub-titles (it is not mandatory, but it will be easy for the readers to understand). Scientists who read this part will understand what they read. However, it can be difficult to understand from the point of view of trainers.
Authors’ Answer: Thank you for the suggestion. The Formative Years is the most complex and important part of the model. We reorganized some paragraphs, added some information (marked in yellow), and created sub-titles (Marked in yellow) to make this part of the text more understandable for all readers.
- Likewise, I suggest you to simplify or make more descriptive parts of "Definitive phase" and "Decision phase". I really had a hard time understanding it while reading it.
Authors’ Answer: Thank you for the suggestion. As we did in Formative Years, we added sub-titles and rewrote some paragraphs to make this part of the text more understandable.
Finally, I recommend that you have the writing language you used in your research edited by a native English language expert. It will become easier to read and will also become simpler.
Authors’ Answer: In addition to the suggested changes in the method, we sent the text to a native speaker who made several changes. We believe that the changes in the writing and the changes made after the reviewers' suggestions helped to make the text clearer and more understandable.

Round 2
Reviewer 1 Report
I have no further comments.
Reviewer 3 Report
Thank you for your efforts. Your manuscript is ready to publish
Best